# Griseofulvin: An Updated Overview of Old and Current Knowledge

**DOI:** 10.3390/molecules27207034

**Published:** 2022-10-18

**Authors:** Parisa Aris, Yulong Wei, Masoud Mohamadzadeh, Xuhua Xia

**Affiliations:** 1Department of Biology, University of Ottawa, 30 Marie Curie, P.O. Box 450, Station A, Ottawa, ON K1N 6N5, Canada; 2Department of Microbial Pathogenesis, Yale University School of Medicine, New Haven, CT 06519, USA; 3Department of Chemistry, Faculty of Sciences, University of Hormozgan, Bandar Abbas 71961, Iran; 4Ottawa Institute of Systems Biology, Ottawa, ON K1H 8M5, Canada

**Keywords:** drug repurposing, dermatophytic fungi, griseofulvin, griseofulvin derivatives, *gsf* gene cluster, polyketide compound, SARS-CoV-2, spindle microtubule

## Abstract

Griseofulvin is an antifungal polyketide metabolite produced mainly by ascomycetes. Since it was commercially introduced in 1959, griseofulvin has been used in treating dermatophyte infections. This fungistatic has gained increasing interest for multifunctional applications in the last decades due to its potential to disrupt mitosis and cell division in human cancer cells and arrest hepatitis C virus replication. In addition to these inhibitory effects, we and others found griseofulvin may enhance ACE2 function, contribute to vascular vasodilation, and improve capillary blood flow. Furthermore, molecular docking analysis revealed that griseofulvin and its derivatives have good binding potential with SARS-CoV-2 main protease, RNA-dependent RNA polymerase (RdRp), and spike protein receptor-binding domain (RBD), suggesting its inhibitory effects on SARS-CoV-2 entry and viral replication. These findings imply the repurposing potentials of the FDA-approved drug griseofulvin in designing and developing novel therapeutic interventions. In this review, we have summarized the available information from its discovery to recent progress in this growing field. Additionally, explored is the possible mechanism leading to rare hepatitis induced by griseofulvin. We found that griseofulvin and its metabolites, including 6-desmethylgriseofulvin (6-DMG) and 4- desmethylgriseofulvin (4-DMG), have favorable interactions with cytokeratin intermediate filament proteins (K8 and K18), ranging from −3.34 to −5.61 kcal mol^−1^. Therefore, they could be responsible for liver injury and Mallory body (MB) formation in hepatocytes of human, mouse, and rat treated with griseofulvin. Moreover, the stronger binding of griseofulvin to K18 in rodents than in human may explain the observed difference in the severity of hepatitis between rodents and human.

## 1. Introduction

Griseofulvin (C_17_H_17_ClO_6_) is a natural product that was first discovered and isolated from *Penicillium griseofulvum* in the 1939. In addition to *Penicillium*, griseofulvin may be isolated from other genera of ascomycetes including *Xylaria flabelliformis*, *Abieticola koreana*, and *Stachybotrys levispora* [1,2,3,4]. Griseofulvin has a wide range of applications, from agriculture to medicine. In agriculture, griseofulvin is used as a crop protectant to prevent fungal colonization and infection [5]. In medicine, griseofulvin has been widely used as an antifungal drug and in treating ringworm and dermatophyte infections in humans and animals [6,7] due to its low toxicity. In cancer research, griseofulvin has shown inhibitory effects on cancer cell division and may induce cell death through interaction with the mitotic spindle microtubule [8]. Furthermore, griseofulvin may inhibit hepatitis C virus replication by interfering with microtubule polymerization in human cells [9]. 

Griseofulvin has long been known to cause vasodilation and improve capillary blood flow [10,11]. A new look at this old drug in our study revealed that griseofulvin has a good binding potential with the SARS-CoV-2 main protease and with the human ACE2 receptor [12]. ACE2 facilitates the breakdown of ANG II which reduce blood pressure and inflammation [13]. Our molecular docking results suggest a promising direction in drug repurposing to enhance ACE2 breakdown to relieve COVID symptoms [12]. 

In this review article, we summarized the history of griseofulvin research since its discovery to the recent findings that suggest a promising future for the clinical application of this fungal secondary metabolite. We also studied the griseofulvin interaction with cytoplasmic intermediate filament proteins using molecular docking to find the binding affinity toward keratin 8 and 18 in human, rat, and mouse. Our findings revealed that griseofulvin has good binding potential with K18 in rat, followed by mouse, which helps explain the significance of mild liver problems in human and the more serious problems in mice.

## 2. Discovery of Griseofulvin and Its Biosynthetic Gene Cluster in Fungi

In 1939, griseofulvin was first isolated and is still widely used as an antifungal mainly caused by dermatophytes [6,7]. Over the next decade of research, Brian and Hemming (1947) discovered a metabolite of *Penicillium janczawski* that could cause hyphal waving of plant pathogen *Botrytis allii* and named it the ‘curling factor’ [14]. This substance was later shown to be biologically and chemically identical to the product of *P. griseofulvum* which was identified and isolated by Oxford et al. (1939) under the name of griseofulvin [15].

Griseofulvin is produced from many species of the genus *Penicillium*, which is one of the most important fungal taxa frequently found around the world [2,3,4,16,17,18,19,20,21,22,23,24,25,26,27]. Following *P. griseofulvum*, *P. lanosocoeruleum*, a plant pathogen, is one of the most significant fungi that has recently been shown to produce griseofulvin [28]. Further, *Abieticola koreana*, *Stachybotrys levispor,* and *Xylaria flabelliformis (previously known as X. cubensis)* were also shown to produce griseofulvin [2,3,4]. Of note, fungal endophytes such as *Xylaria* are abundant in plant tissue that produce griseofulvin as an antifungal against plant pathogenic fungi [1,29,30].

Griseofulvin is a polyketide derived from acetyl and malonyl CoA precursor molecules to form dehydrogriseofulvin. The A-B-C ring structures of griseofulvin were first discovered by oxidative degradation [31]. Findings of Brich et al. (1958) showed that 6-methylsalicylic acid (6-MSA), the carbon skeleton of griseofulvin, is assembled from one acetyl CoA and three molecules of malonyl CoA [32,33]. The biosynthetic structure of griseofulvin was elucidated by ^13^C-NMR study using singly and doubly labeled acetate [34], and the crystal structure of griseofulvin was revealed by X-ray crystallography and UV spectroscopy [35,36,37].

In 2010, the griseofulvin biosynthetic gene cluster (*gsf* BGC) responsible for griseofulvin production in *P. aethiopicum* was fully discovered through shotgun sequencing and bioinformatics mining (Figure 1) [16]. Recently, we performed a comparative genome study of more than 266 fungal species and showed that *Memnoniella echinate*, with only seven out of the thirteen putative *gsf* cluster genes (Figure 1), can still synthesize Griseofulvin [38]. A phylogenetic analysis reconstructed with 18S rRNA sequences shows the global relationships among whole *gsf* BGCs of 13 surveyed fungal species including *P. aethiopicum*, *X. flabelliformis*, *M. echinata*, *P. griseofulvum*, *P. vulpinum*, *P. coprophilum*, and *A. alliaceus* (Figure 1). The 18S rRNA sequences do not have strain information, so there is no one to one correspondence between the leaves and the gene clusters when multiple strains are involved.

## 3. Biosynthesis of Griseofulvin

### 3.1. Gene Cluster Evolution Hypothesis 

Genes producing secondary metabolites in fungi are frequently linked and located in biosynthetic gene clusters (BGCs) [39]. It has been hypothesized that horizontal gene transfer (HGT) plays a vital role in this phenomenon. This is so because genes involved in fungal metabolic pathways are frequently physically clustered, increasing the possibility of pathway transfers occurring in a single event. There is evidence that sterigmatocystin (ST), a highly toxic secondary metabolite that serves as a precursor to aflatoxins (AF), was horizontally transferred from Aspergillus [40]. However, it is believed that the metabolic gene clusters may have evolved from bacteria as almost half of the bacterial genes are located in an operon, a group of genes under the control of a promoter, and it may be essential to control the expression of downstream genes [41]. Further investigations have revealed that the gene clusters are usually located at the end of the chromosomes and flanked by mobile genetic elements (MGEs) that facilitate gene transfer within genomes and between species [41,42,43].

### 3.2. Griseofulvin Biosynthetic Gene Cluster

In 2010, the putative griseofulvin gene cluster containing the PKS genes *gsfA*–*gsfK*, *gsfR1*, and *gsfR2* was first identified through shotgun sequencing and bioinformatic mining in *P. aethiopicum* [16]. Later, single-gene deletion and biochemical test procedures were used to investigate the role of these *gsf* genes in griseofulvin biosynthesis [44], with the key findings summarized in Figure 2. GsfA initiates the biosynthesis of griseofulvin by combining one acetyl CoA and six malonyl CoA molecules to generate heptaketide backbone benzophenone 5a. GsfA is a nonreducing polyketide synthase (NR-PKS) that consists of several functional domains, including starter unit ACP transacylase (SAT), ketosynthase (KS), malonyl-CoA-ACP transacylase (AT), product template (PT), and an acyl carrier protein (ACP). The GsfA PT domain may mediate the cyclization of aromatic rings to form the intermediate benzophenone. The intermediate griseophenone C is produced from the methylation of phenols on benzophenone 5a by the O-methyltransferases (*gsfB* and *gsfC*). Following, griseophenone C is chlorinated to convert into griseophenone B by the halogenase *gsfI*, and the grisan core is produced from griseophenone B by the phenol oxidative activity of *gsf F*. The grisan core is finally transformed into griseofulvin by two further processes, methylation at 5-OH mediated by *gsf D* and enoyl redcution catalyzed by *gsf* E (Figure 2) [44].

Not all *gsf* genes are essential for griseofulvin biosynthesis and function. For instance, the deletion of *gsfA* or *gsfI* disrupts the production of griseofulvin since both proteins are essential for the *gsf* biosynthetic pathway [16,44,45]. Additionally, the deletion of *gsfK* had no impact on biosynthesis in *P. aethiopicum* [44]. Further, when compared to *P. aethiopicum*, *P. griseofulvum* lacks *gsfK*, *gsfH*, and *gsfR2* in the *gsf* BGC [45].

We recently conducted a comparative genome study in which the *gsf* BGC of 12 fungal genomes was identified, annotated, and compared. In this study, in addition to the *Penecillium* species, *gsf* BGC was found for the first time in *Aspergillus alliaceus*, *Aspergillus burnettii*, and *Memnoniella echinata*. Overall, our findings suggest that a total of seven genes (*gsfA*–*gsfF*, *gsfI*) out of the thirteen in *P. aethiopicum* were conserved by the majority of the genomes investigated and no gene rearrangements were reported at the *gsf* BGC [38].

### 3.3. Genetic Regulation of Griseofulvin Biosynthesis

The *gsfR1* and *gsfR2* encode for putative transcription factors in the griseofulvin gene cluster. However, their functions in the *gsf* gene cluster remain debatable. The *gsfR1* gene may not only impact griseofulvin biosynthesis but also serves as a critical regulator of other secondary metabolisms [46], such as patulin produced by *P. griseofulvum* [45]. Indeed, the *gsfR1* gene deletion increased griseofulvin biosynthesis by increasing *gsfA* gene expression when tested on apples and PDA medium in vitro, suggesting *gsfR1* as a negative regulator of the cluster. However, in different culture conditions, such as high concentrations of nitrogen and complex sugars, and on media supplemented with peptone, the *gsfR1* positively regulates griseofulvin biosynthesis [46]. These results indicate that *gsfR1* can respond differently depending on external stimulation, particularly nitrogen and carbon availability. Surprisingly, the gene *gsfR1* was missing in the recently released genome of griseofulvin producer *Xylaria flabelliformis* [2], indicating that griseofulvin biosynthesis could be regulated independently of *gsfR1* action.

It is noteworthy that the *gsfR2* gene in *P. griseofulvum* was found in a separate genomic region of the biosynthetic gene cluster, in contrast to *P. aethiopicum* [45] and *gsfR2* deletion did not affect griseofulvin biosynthesis, suggesting that this gene is probably involved in a different pathway or another secondary metabolite biosynthesis [46].

### 3.4. Environmental Factors in Producing Griseofulvin

The gene cluster expression is mostly synchronized in response to environmental conditions. Several factors appear to influence producing griseofulvin, of which organic compounds are one of the most critical environmental components [47]. The yield of griseofulvin increases in rich media with glucose, acetate, and succinate. Of note is that a nitrogen concentration of less than 0.04 g% or more than 0.4 g% may inhibit the griseofulvin biosynthesis [48]. Furthermore, the ATP/ADP ratio is a significant parameter of cellular energy status that regulates griseofulvin synthesis; the more the energy level, the more griseofulvin production [49,50]. Another important factor is pH. The highest level of griseofulvin yields in a pH ranging from 5.5 to 6 [51].

Secondary metabolites are mainly synthesized during the stationary-phase growth by microorganisms to improve their ability to thrive in diverse environments [52,53]. It may be assumed that the advantages of griseofulvin production are to defend against other fungal species and reduce competition. Even though only species that encounter such competition would preserve the griseofulvin gene cluster in their genomes, given the cost of time and energy. Thus, when the beneficial effects of producing griseofulvin outweigh the energy cost of biosynthesis, the organism would have increased fitness, according to the natural selection theory [54,55]. One example is the fungal endophyte, *Xylaria flabelliformis*, which is commonly found within plant tissue and the synthesis of griseofulvin as an antifungal against plant pathogenic fungi [2].

## 4. Physiochemical Properties of Griseofulvin

Griseofulvin is a fungal secondary polyketide metabolite that is soluble in ethanol and methanol but has poor solubility in water [6]. One notable feature of griseofulvin is its ability to tolerate heat stress and maintain its function at a high temperature of 121 °C without losing functional properties [47]. These properties are important to understand their medical applications because they affect absorption, transportation, excretion and degradation.

## 5. Medical Applications

### 5.1. Antifungal Application 

The first use of griseofulvin in medicine can be traced back to treating ringworm infections caused by *Microsporum canis* in guinea pigs [56]. Further investigations revealed its effect on not only the genus *Microsporum* but *Trichophyton* and *Epidermophyton* [57]. Griseofulvin was approved by Food and Drug Administration (FDA) in 1959, and it is still in use today and is considered one of the most widely used treatments for dermatophytes fungal infections in humans and animals.

### 5.2. Non-fungal Inflammatory Diseases

Further studies reported the effectiveness of griseofulvin in the treatment of non-fungal skin inflammation diseases, including lichen planus [58] and chronic purpuric dermatosis [59], suggesting that it may also have anti-inflammatory and immunomodulatory potential. This is while, earlier studies have shown that griseofulvin is effective in treating of the shoulder-hand syndrome [60], and a few other inflammatory, rheumatic conditions, such as posttraumatic reflex dystrophies and scapulohumeral periarthritis [61].

### 5.3. Cardiovascular Applications

It has been shown that intravenous administration of griseofulvin improves coronary blood flow and decreases blood pressure, indicating its peripheral vasodilation effects. In addition, griseofulvin increases myocardial heart rate by directly acting on the myocardium and vascular smooth muscle rather than through central nervous systems or humoral mechanisms [10,11].

### 5.4. Antitumor Applications

Griseofulvin has gained attention for its potential application in cancer chemotherapy because of its low toxicity. In tumor cell lines, antifungal drug griseofulvin inhibits tumor growth and several forms of cancer cell proliferation by suppressing spindle microtubule dynamics, mitotic arrest, and cell death in multipolar spindles, but not in fibroblasts and keratinocytes containing normal centrosome composition. Indeed, griseofulvin binds to the αβ intra-dimer tubulin interface and induces mitotic arrest by a variety of mitotic abnormalities, such as misaligned chromosomes and multipolar spindles, resulting in cells containing fragmented nuclei. These cells exhibited increased p53 accumulation in the nucleus, indicating that the cells undergo apoptosis [8,62].

### 5.5. Antiviral Applications 

Griseofulvin was used for zoster-associated pain relief and prevented the development of further lesions within two days of starting treatment [63]. Further studies have shown that griseofulvin suppresses the hepatitis C virus (HCV) replication by arresting the human cell cycle at the G2/M phase and acting on microtubule polymerization but does not affect HCV internal ribosome entry site (IRES) dependent translation. A synergistic inhibitory effect of griseofulvin and interferon alpha (IFNα) was also noted in Huh7/Rep-Feo (hepatoma cell line containing HCV 1b replicons that expresses a chimeric protein consisting of neomycin phosphotransferase and firefly luciferase) cells [9].

## 6. Pharmacokinetics of Griseofulvin

Griseofulvin is commonly used orally, although poor intestinal absorption may lead to failure in treatment. A spectrophotofluorometric study showed that griseofulvin absorption and effectiveness are dramatically increased with the high-fat diet [64]. Indeed, the oral bioavailability of griseofulvin-loaded liposomes may be enhanced significantly by high drug encapsulation efficiency and small liposome size [65].

Griseofulvin absorption is greatest from the duodenum and least from the stomach, but colon absorption is hardly noticeable [66]. In addition, a higher level of griseofulvin was found in the lung after intravenous administration, while the liver had a significantly higher level after oral administration [67]. Furthermore, griseofulvin can be secreted through sweat glands and appears in the keratinized layer within four hours after a single-dose administration to inhibit fungal growth [68]. The treatment response time depends on the keratin thickness at the injection site. Accordingly, healing fungal toe-nail infections may take longer than hair or skin infections [69].

The biological half-life of griseofulvin is 9 to 21 hours in the blood [70], and it is excreted as 6-desmethylgriseofulvin (6-DMG) and 4- desmethylgriseofulvin (4-DMG) metabolites in urine and feces after being metabolized by the liver microsomal enzyme system [66].

## 7. Characterized Interactions between Griseofulvin and Cellular Components

Griseofulvin enters the dermatophyte through energy-dependent transport processes, binds to the fungal microtubules, interfering the microtubule function, thus inhibiting mitosis [71]. The primary mechanism by which griseofulvin inhibits mitosis at metaphase is to suppress spindle microtubule (MT) dynamics by acting directly at the plus end to increase the stability and suppress the shortening rate at the MT plus ends [72]. Griseofulvin has two possible binding sites in tubulin based on docking studies; one of which overlaps with the Taxol, a chemotherapy medication to treat cancer, the binding site at the beta-tubulin H6-H7 hoop, and the other is located at the interface of αβ tubulin [64]. It has been shown that griseofulvin at a concentration of 30 to 60 μM induced both mitotic G2/M arrest and apoptosis in a human malignant cell line (HL-60) using activation of NF-κB pathway with cell division cycle protein (cdc)2 activation and phosphorylation of B-cell lymphoma (Bcl)2 proteins that regulate cell death [73]. Notably, the β-tubulin gene expression has been reported to decrease after griseofulvin treatment in *T. rubrum* [74].

Further studies have shown that it can be considered a potential therapeutic option for colon and breast cancer by an inhibitory effect on centrosomal clustering activity by altering the interphase microtubule stability. The findings strongly argued that mitotic irregularities and nuclear localization of p53 in human breast cancer MCF-7 cells are induced by kinetic inhibition of microtubule dynamics [8,62]. Likewise, griseofulvin inhibits hepatitis C virus replication by disrupting microtubule polymerization and interrupting the G2/M phase in human cells [9].

Additionally, griseofulvin has been shown effective for inflammatory skin diseases through inducing inhibitory effects on vascular cell adhesion molecule 1 (VCAM-1) in both TNF-alpha and IL-1 stimulated human dermal microvascular endothelial cells (HDMEC) [75].

## 8. Changes in Keratins 8 and 18 in Griseofulvin-Induced Liver Injury

Because of the high concentration of griseofulvin in the liver as the major detoxification site, it has also been reported to cause occasional hepatitis [76,77]. While griseofulvin-caused hepatitis is extremely rare in human, Mallory bodies (MBs) in hepatocytes associated with griseofulvin treatment were frequently reported in rodents [78,79]. The intermediate filament proteins (IFs), such as cytoplasmic keratins 8 and 18 (K8/18), are one of the most important components of a Mallory body [80]. These intermediate filament proteins are believed to play an important role in hepatocyte protection against toxic or mechanical stress, and regulation of cell response to injuries, cell growth, and death [81,82]. K8 and 18 are present in a 1:1 ratio under normal circumstances [83], but liver diseases such as alcoholic hepatitis, hepatic porphyria, and copper metabolism increase the K8/K18 ratio to more than 1, consequently forming the K8/18-containing aggregates known as Mallory bodies, which impairs centrosome and microtubule network function and results in cell death [84,85,86,87]. Likewise, treatment of mice with griseofulvin-containing diet induces the Mallory body formation likely caused by overexpression of Krt8 (Keratin 8) gene [88], and griseofulvin binding to K8 and K18 in hepatocytes, which result in changing the keratin solubility, keratin phosphorylation on specific sites, and causing protein misfolding [81]. Misfolded proteins are refolded with help of cellular chaperone proteins or tagged by ubiquitin for degradation [89]. These findings suggest the binding potential of griseofulvin with cytoskeletons, especially keratin intermediate filaments proteins K8, and K18.

Here, we studied the griseofulvin interaction with K8 and K18 in human, rat, and mouse using molecular docking (Table 1), conducted in AutoDock 4 [90]. The 3D protein structures of the K8 and K18 were downloaded from the RCSB Protein Data Bank (PDB) (https://www.rcsb.org/) or AlphaFold database (https://alphafold.ebi.ac.uk/). The interactions between the receptors and the griseofulvin were visualized using Discovery Studio Visualizer v.20 (https://discover.3ds.com/discovery-studio-visualizer-download). Keratin 8 (Krt8) and keratin 18 (Krt18) genes reside on chromosome 12 (GenBank accession NC_000012) in human, chromosome 15 (NC_000081) in mouse (Mus musculus), and chromosome 7 (NC_051342) in rat (Rattus norvegicus). The Krt8 sequences, after alignment with MAFFT, have 493 aligned sites of which human and mouse K8 differ by 44 aa (89% identity, 86% similarity), human and rat K8 differ by 44 aa (89% identity, 86% similarity) and mouse and rat K8 differ by 27 aa (94% identity, 90% similarity). The alignment of the three Krt18 amino acid sequences have 432 sites, of which human and mouse K18 differ by 48 aa (86% identity, 90% similarity), human and rat K18 differ by 57 aa (84% identity, 89% similarity) and mouse and rat K18 differ by 16 aa (96% identity, 96% similarity). The molecular docking results revealed the K8 residues of Ile 63, Gln 85, and Lys 92 form H bonds with the oxyl groups of griseofulvin, while the K18 residues of Arg 117, Gln 243, Gln 350, Thr 353, and Arg 404 are involved in H-bond interactions (Figure 3). The griseofulvin illustrated low binding energy toward the K8, ranging from −3.17 to −4.6 kcal mol^−1^. However, a higher binding energy toward K18 was observed in rat and mouse with −5.23 and −5.54 kcal mol^−1^ respectively (Table 1). In conclusion, the best griseofulvin interaction was found with Keratin 18 in rat, followed by mouse, based on the lowest binding energy and highest number of hydrogen bonds. This differential binding to K8 and K18 may disrupt the 1:1 ratio between the two and contribute to liver problems. The stronger binding of griseofulvin to K18 in rodents than in human may also explain the observed difference in the severity of hepatitis between rodents and humans.

Amiodarone (Am), an antiarrhythmic medication used to treat and prevent a number of types of cardiac dysrhythmias [91], was used as positive control in this study, because it could be associated with liver injury and Mallory body formation in hepatocytes [92,93]. The molecular docking analysis showed that Arg90 is the only residue that can form hydrogen bonds in the binding cavity of K18, while the docking pose of amiodarone with K8 revealed three hydrogen bonds interactions with amino acids Gly 61 and Ile 63 in human (Table 2). Amiodarone showed the lowest free energy values (−4.68 and −4.85 kcal mol^−1^), corresponding to the high affinity binding site, for human K18. However, the amiodarone exhibited higher binding affinities, with binding energies of −4.14 and −3.64 kcal mol^−1^, to K8 than K18 of mouse and rat respectively.

We also studied the molecular interactions of griseofulvin metabolites, including 6-desmethylgriseofulvin (6-DMG) and 4- desmethylgriseofulvin (4-DMG) [66], with K8 and K18 in human, mouse, and rat. Our in silico analysis exhibited slightly higher affinities for 6-DMG and 4-DMG metabolites than griseofulvin to mouse and rat K18. In addition, the number of hydrophobic and hydrogen bonds between griseofulvin metabolites and cytokeratin proteins (K8 and K18) were greater than griseofulvin in human, mouse, and rat. These findings suggest that griseofulvin metabolites have favorable interactions with keratins 8 and 18 (K8 and K18), and therefore they could be responsible for liver injury and Mallory body formation in hepatocytes of human and rodents treated with griseofulvin. The molecular docking details of 4-DMG and 6-DMG with K8 and K18 are shown in Table 3 and Table 4 respectively.

## 9. Repurposing of Griseofulvin

Computational drug repurposing accelerates drug development and reduces the time, effort, and expenses associated with the drug discovery process [94,95]. In our recent study, griseofulvin and its derivatives were repurposed as potent inhibitors of SARS-CoV-2 entry and viral replication. The inhibitory effects of griseofulvin and its derivatives on essential SARS-CoV-2 main protease, RNA-dependent RNA polymerase (RdRp), spike protein receptor-binding domain (RBD), and angiotensin-converting enzyme 2 (ACE2) from humans were investigated using molecular docking and molecular dynamics simulation. Our findings showed the activity of these compounds against SARS-CoV-2 target proteins. The highest docking score with the most hydrogen bond interactions to COVID-19 main protease was observed for griseofulvin. However, griseofulvin was also shown to interact favorably with ACE2, RBD, and RdRb proteins [12].

In addition to these inhibitory effects, griseofulvin may enhance ACE2 function to regulate the renin-angiotensin-aldosterone system (RAAS). ACE converts angiotensin I (ANG I) to angiotensin II (ANG II), which increases blood pressure and causes inflammation. ACE2 facilitates the breakdown of ANG II which lowers blood pressure and inflammation while maintaining homeostasis and preventing tissue damage [13]. Inhibition of ACE2 function caused by SARS-CoV-2 binding to the ACE2 receptor leads to an increase in ANG II protein and its deleterious effects. It is well-known that griseofulvin has vasodilation and increased capillary blood flow effects [10,11], although the molecular mechanism by which griseofulvin affects blood pressure remains unclear. Results from our molecular docking study put forward the hypothesis that griseofulvin binds to ACE2 and competes against SARS-CoV-2 binding to the extracellular domain of ACE2, therefore improving ACE2 vasodilation and tissue protection functions [12]. These findings suggest that griseofulvin and its derivatives might be considered when developing future SARS-CoV-2 treatment options.

## 10. Conclusions

Since the discovery of the FDA-approved drug griseofulvin, practical clinical information and applications have been well established in the literature. Nevertheless, computational and bioinformatics analysis have promoted a revolution in scientific discovery, in which repurposing and developing new indications for already approved drugs gained tremendous interest during past years. Griseofulvin has proven to be a safe therapeutic agent in managing and suppressing dermatophyte infections, however its potential to manage cancer, hepatitis C, and SARS-CoV-2 have been predicted in theory and are under experimental investigations. Although it is well documented that griseofulvin interacts with spindle microtubules, our molecular docking study showed its binding potential toward keratin intermediate filament proteins in human, rat, and mouse, which explains more serious liver problems in rodents than humans. Here, we also reviewed the earliest to the most current knowledge about griseofulvin from a medical perspective. Further studies on griseofulvin may reveal new applications, design and development in future therapeutic interventions.

## Figures and Tables

**Figure 1 molecules-27-07034-f001:**
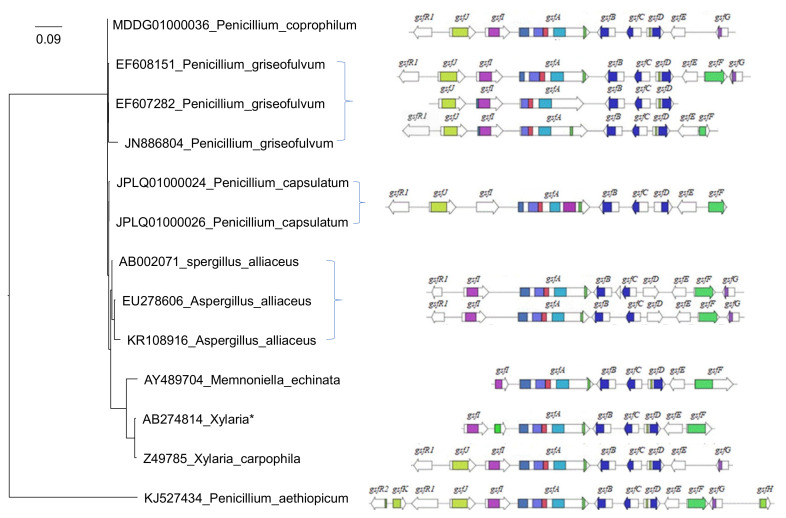
Griseofulvin biosynthetic gene cluster in *P. aethiopicum* and other fungal species The arrows show the *gsf* gene candidates and their orientations in griseofulvin biosynthesis gene cluster. The phylogenetic tree is reconstructed with 18S rRNA sequences (in the format of accession_species) with the PhyML software after alignment with MAFFT. The *gsf* genes for *Xlaria** is *X. flabelliformis* which does not have an 18S rRNA sequence. *X. angulosa* was used instead. *P. aethiopicum* has only 33 nt of the 18S rRNA in contrast to other sequences that are ~1700 nt long.

**Figure 2 molecules-27-07034-f002:**
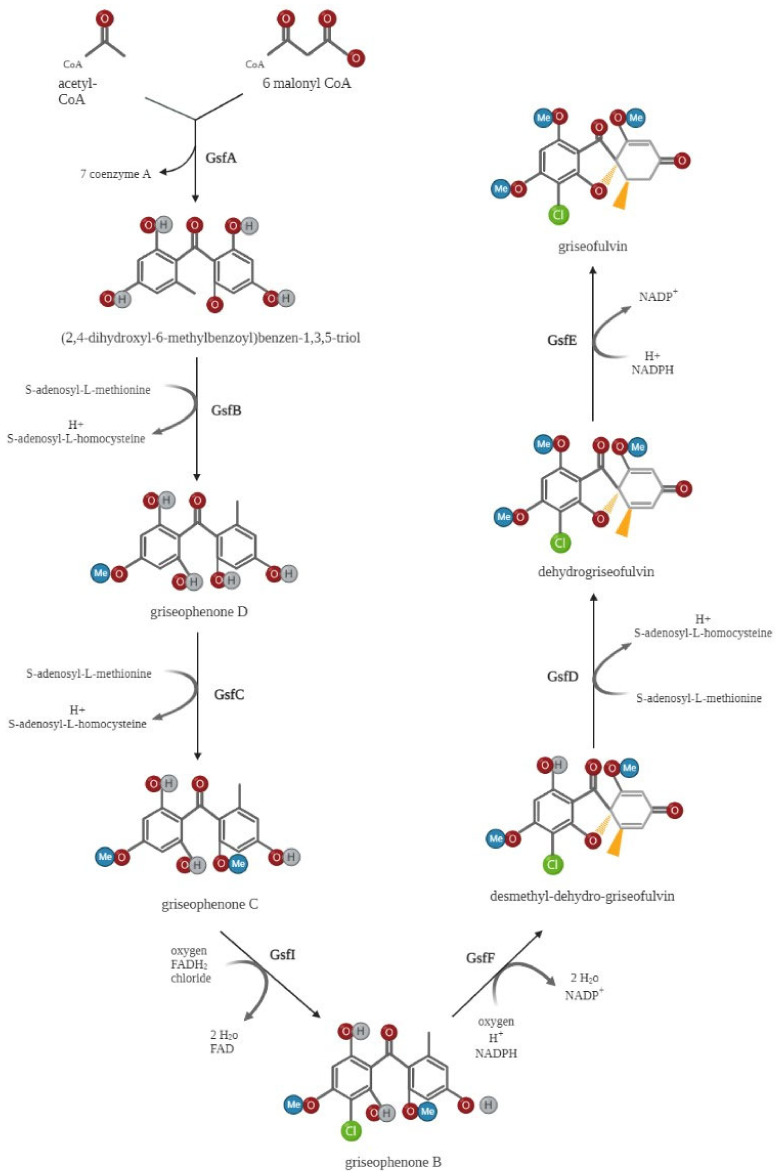
Griseofulvin biosynthetic pathway. The nonreducing polyketide synthase *gsfA* initiates the synthesis of griseofulvin by combining one acetyl-CoA and six malonyl-CoA units to form the heptaketide backbone of benzophenone 5a. The O-methyltransferases (*gsfB* and *gsfC*) then methylate phenols on benzophenone 5a to generate the intermediate griseophenone C. Following this process, griseophenone C is chlorinated by the halogenase gsfI to produce griseophenone B, and griseophenone B is then converted to the grisan core by phenol oxidative activity of *gsfF*. The grisan core is ultimately converted into griseofulvin by two further processes: enoyl reduction catalyzed by *gsfE* and methylation at 5-OH mediated by *gsfD*. Created by Biorender.com.

**Figure 3 molecules-27-07034-f003:**
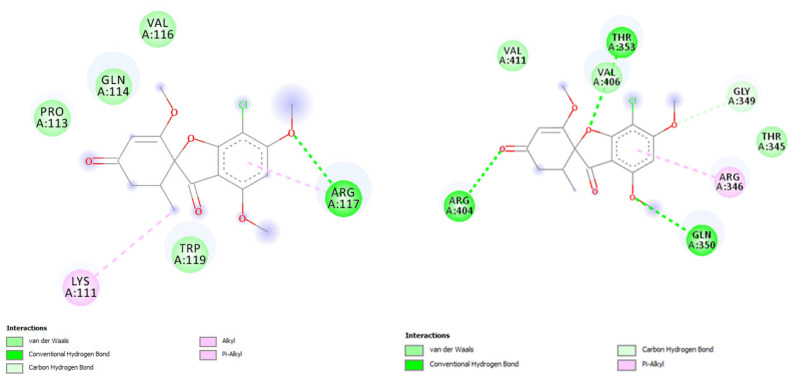
2D binding model of griseofulvin docked into the active site of keratin 18 (K18) in mouse (**left**) and rat (**right**). The dotted lines show the key interacting residues of the K18 toward griseofulvin as a ligand. The green, light green, and pink dotted lines show conventional hydrogen bonds, van der Waals, alkyl or Pi-alkyl interactions, respectively.

**Table 1 molecules-27-07034-t001:** Molecular docking details of griseofulvin interaction with keratin 8 (K8), and keratin 18 (K18) in human, rat, and mouse.

Keratin	Species	PDB or Alpha Fold IDs	ΔG_binding_ (Kcal/mol)	Hydrogen Bond	Hydrophobic Interaction
K8	Human	3K3C	−3.17	ILE 63	GLY 61, GLY 62, THR 64
		7K3X	−3.2	ILE 63	GLY 61
		7K3Y	−3.61	-	SER 58
		P05787F1	−4.6	GLN 85, LYS 92	ARG 88, LYS 92
	Mouse	P11679F1	−4.1	-	LYS 420, THR 421
	Rat	Q10758F1	−4.55	-	SER 15, GLY 16, PRO 17
K18	Human	A0A024RAY2F1	−4.72	-	LEU 87, ARG 90, SER 93, TYR94
		P05783F1	−4.99	GLN 243	SER 242, GLN 243, ASP 244, LYS 247, ILE 248, ASP 251
	Mouse	P05784F1	−5.54	ARG 117	LYS 111, ARG 117
	Rat	Q5BJY9F1	−5.23	GLN 350, THR 353, ARG 404	ARG 346, GLY 349

**Table 2 molecules-27-07034-t002:** Molecular docking details of Amiodarone interaction with Keratin 8 (K8), and Keratin 18 (K18) in Human, Rat, and Mouse.

Keratin	Species	PDB or Alpha Fold IDs	ΔG_binding_ (Kcal/mol)	Hydrogen Bond	Hydrophobic Interaction
K8	Human	3K3C	−3.38	GLY 61, ILE 63	MET 60, GLY 61, ILE 63
		7K3X	>0	-	-
		7K3Y	>0	-	-
		P05787F1	−4.1	-	MET 406, SER 410, HIS 412
	Mouse	P11679F1	-4.14	-	MET 412, MET 415, ILE 417
	Rat	Q10758F1	−3.64	ARG 49	SER 51, LEU 52, PHE 55
K18	Human	A0A024RAY2F1	−4.69	ARG 90	LEU 87, ARG 90, LEU 91, TYR 94
		P05783F1	−4.85	-	LEU 87, ARG 90, LEU 91, TYR 94
	Mouse	P05784F1	−2.51	-	GLU 414
	Rat	Q5BJY9F1	−0.21	-	ARG 374, GLU 378

**Table 3 molecules-27-07034-t003:** Molecular docking details of 4-desmethylgriseofulvin (4-DMG) interaction with Keratin 8 (K8), and Keratin 18 (K18) in Human, Rat, and Mouse.

Keratin	Species	PDB or Alpha Fold IDs	ΔG_binding_ (Kcal/mol)	Hydrogen Bond	Hydrophobic Interaction
K8	Human	3K3C	−3.34	GLY 62, THR 64	GLY 61, GLY 62
		7K3X	−3.34	GLY 61, ILE 63	GLY 61, ILE 63
		7K3Y	−3.8	SER 58	SER 58, ALA 57
		P05787F1	−5.43	THR 64, ALA 65	THR 64, ALA 65,VAL 66, THR 67
	Mouse	P11679F1	−4.14	THR 419, LYS 420, THR 421	THR 421, THR 422
	Rat	Q10758F1	−3.93	SER 21, ARG 23	ALA 19, PHE 20, SER 21
K18	Human	A0A024RAY2F1	−4.48	ARG 90	ARG 90
		P05783F1	−4.96	SER 242, ASP 244	GLN 243, LYS 247, ASP 251
	Mouse	P05784F1	−5.28	GLN 114, ARG 117	LYS 111, PRO 113, GLN 114, ARG 117
	Rat	Q5BJY9F1	−5.22	THR 353, ARG 404	ARG 346, GLN 350, VAL 411

**Table 4 molecules-27-07034-t004:** Molecular docking details of 6-desmethylgriseofulvin (6-DMG) interaction with Keratin 8 (K8), and Keratin 18 (K18) in Human, Rat, and Mouse.

Keratin	Species	PDB or Alpha Fold IDs	ΔG_binding_ (Kcal/mol)	Hydrogen Bond	Hydrophobic Interaction
K8	Human	3K3C	−4.04	GLY 62, THR 64	GLY 61, GLY 62
		7K3X	−3.71	ILE 63	ILE 63
		7K3Y	−3.95	GLY 56, SER 58	GLY 56, SER 58
		P05787F1	−4.85	LEU 72	LEU 72, PRO 75, VAL 77
	Mouse	P11679F1	−4.29	GLN 134, SER 138	LEU 131, GLN 134, GLN 135, SER 138
	Rat	Q10758F1	−4.50	SER 21	ARG 18, ALA 19
K18	Human	A0A024RAY2F1	−4.88	TYR94	LEU 91, TYR 94
		P05783F1	−4.81	GLU 236	SER230, SER 231, LEU 233, VAL 235, GLU 236
	Mouse	P05784F1	−5.61	LEU 108, TPR 119	LYS 111, PRO 113, VAL 116
	Rat	Q5BJY9F1	−5.33	ARG 346, GLN 350, ARG 404	ARG 346, GLN 350, VAL 411

## Data Availability

Not applicable.

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
