# Peer review of "Griseofulvin: An Updated Overview of Old and Current Knowledge"

_molecules, 2022, doi:10.3390/molecules27207034_

Round 1

Reviewer 1 Report

The manuscript is very well written. It does not contain substantive or logical errors.

Author Response

Reviwer1:

The manuscript is very well written. It does not contain substantive or logical errors.

Thank you so much!  

Reviewer 2 Report

The biosynthetic structure of griseofulvin was elucidated by 13C-NMR study using singly and doubly labelled acetate(Simpson and Holker, 1977),

Add a space between acetate and the parenthesis

Figure 3 is described in the paper but it wasn’t showed, add figure 3

Author Response

Reviwer2:

  1. The biosynthetic structure of griseofulvin was elucidated by 13C-NMR study using singly and doubly labelled acetate(Simpson and Holker, 1977), Add a space between acetate and the parenthesis

A space between acetate and the parenthesis was added to the line 82.

  1. Figure 3 is described in the paper but it wasn’t showed, add figure 3

 The figure 3 is added to the page 15.

Reviewer 3 Report

Dear authors,

Drug rehabilitation is a process that supports therapeutic options for a number of diseases. The proposed review is an interesting perspective on a long-known antifungal drug. The new possibilities for its application in various fields of medicine are intriguing and deserve attention. Thus, I consider the proposed article suitable for publication after major improvements.

I have a few recommendations. I have also pointed out to the authors some inaccuracies and omissions, as follows.

·         I do not agree with the statement of the authors from the Introduction that "griseofulvin has been widely used as an anti-fungal drug and in treating ringworm and dermatophyte infections”. The drug is not a broad-spectrum antifungal agent, but has limited use in infections caused by Trichophyton and Microsporum (https://go.drugbank.com/drugs/DB00400)

·         Also in the Introduction, data from the authors' previous research is included, but no reference is indicated

·         In point 4, in order for the pharmacokinetic characterization to be complete, data on the excretion of the drug should also be included. I suggest to the authors that it be renamed Pharmacokinetics of Griseofulvin

·         In point 5, in my opinion, the main mechanism of action of the substance as an antifungal pharmacotherapeutic agent should be indicated at the beginning. And then to describe the details at the molecular level, as well as the experimental data and hypotheses that follow them. The last sentence is unclear to the reader of the text, I recommend rewording it.

·         I suggest that item 7 should be moved before the pharmacokinetic characteristics of the drug (item 4) to follow the logic in the drug descriptions. In item 7.4. the name is misspelled.

·         Points 8.1. to 8.4, inclusive, should be moved after item 2 and before the physico-chemical characteristics of the substance, because they describe its biosynthesis and the logic of the exposition is that this information should follow that of item 2 "Discovery of Griseofulvin and its biosynthetic gene cluster in fungi"

·         In my opinion, item 8.5. does not contribute to the review, which considers griseofulvin as a substance with bigger therapeutic potential. I suggest to the authors that it be excluded from the text.

·         Figure 2 should be moved closer to the first mention in the text. Now it's right below the mention of Figure 3, which is missing altogether - I can only find its title.

·         On line 3 of point 9, own results are again cited without specifying a reference.

·         In Conclusion: “Griseofulvin has proven to be a safe and multipo-tent therapeutic agent in many agricultural and medical applications in managing and suppressing dermatophyte infections, cancer, hepatitis C, and SARS-CoV-2 virus”. The substance is currently applied as an antifungal drug, and the other biological activities that are indicated in this sentence are in the experimental or hypothetical phase. It is not correct to list them as an established therapeutic application.

·         Also, I suggest the authors to exclude from the Conclusion the sentence about griseofulvin derivatives.

·         Overall, I do not get the impression that this review offers a clinical update. In my opinion, it includes an updated summary of griseofulvin - pharmacological characterization and therapeutic application, factors determining its biosynthesis, as well as new therapeutic opportunities. Clinical data for the latter are not provided, and the conclusions drawn are based on docking and experimental data. I recommend revising the title.

Author Response

Reviwer3:

Dear authors,

Drug rehabilitation is a process that supports therapeutic options for a number of diseases. The proposed review is an interesting perspective on a long-known antifungal drug. The new possibilities for its application in various fields of medicine are intriguing and deserve attention. Thus, I consider the proposed article suitable for publication after major improvements.

I have a few recommendations. I have also pointed out to the authors some inaccuracies and omissions, as follows.

  1. I do not agree with the statement of the authors from the Introduction that "griseofulvin has been widely used as an anti-fungal drug and in treating ringworm and dermatophyte infections”. The drug is not a broad-spectrum antifungal agent, but has limited use in infections caused by Trichophyton and Microsporum (https://go.drugbank.com/drugs/DB00400)

 This sentence is revised in line 14. The “widely” is removed.

  1. Also in the Introduction, data from the authors' previous research is included, but no reference is indicated

 The reference is added to the line 45.

  1. In point 4, in order for the pharmacokinetic characterization to be complete, data on the excretion of the drug should also be included. I suggest to the authors that it be renamed “Pharmacokinetics of Griseofulvin”

 The data on the excretion of the drug was explained in lines 284-7. This point renamed to “Pharmacokinetics of Griseofulvin”.

  1. In point 5, in my opinion, the main mechanism of action of the substance as an antifungal pharmacotherapeutic agent should be indicated at the beginning. And then to describe the details at the molecular level, as well as the experimental data and hypotheses that follow them. The last sentence is unclear to the reader of the text, I recommend rewording it.

The main mechanism of action of the antifungal pharmacotherapeutic agent is added to the lines 289-91.

 The last sentence is revised in lines 310-2.

  1. I suggest that item 7 should be moved before the pharmacokinetic characteristics of the drug (item 4) to follow the logic in the drug descriptions. In item 7.4. the name is misspelled.

Item 7 is moved to the lines 228-68. The misspelled name (griseofulvin) is corrected in line 251.

  1. Points 8.1. to 8.4, inclusive, should be moved after item 2 and before the physico-chemical characteristics of the substance, because they describe its biosynthesis and the logic of the exposition is that this information should follow that of item 2 "Discovery of Griseofulvin and its biosynthetic gene cluster in fungi"

 The points 8.1 to 8.4 (Biosynthesis of Griseofulvin) are moved to the lines 107-220 after item 2.

  1. In my opinion, item 8.5. does not contribute to the review, which considers griseofulvin as a substance with bigger therapeutic potential. I suggest to the authors that it be excluded from the text.

The item 8.5 (Derivatives of Griseofulvin) is excluded from the text.

  1. Figure 2 should be moved closer to the first mention in the text. Now it's right below the mention of Figure 3, which is missing altogether - I can only find its title.

Figure 2 is moved closer to its first mention and now it is located in page 7. 

  1. On line 3 of point 9, own results are again cited without specifying a reference.

 The citation is added to the line 382.

  1. In Conclusion: “Griseofulvin has proven to be a safe and multipo-tent therapeutic agent in many agricultural and medical applications in managing and suppressing dermatophyte infections, cancer, hepatitis C, and SARS-CoV-2 virus”. The substance is currently applied as an antifungal drug, and the other biological activities that are indicated in this sentence are in the experimental or hypothetical phase. It is not correct to list them as an established therapeutic application.

The sentence in conclusion is reworded in lines 401-4.

  1. Also, I suggest the authors to exclude from the Conclusion the sentence about griseofulvin derivatives.

 The sentence about griseofulvin derivatives is excluded from the text in lines 404-5.

  1. Overall, I do not get the impression that this review offers a clinical update. In my opinion, it includes an updated summary of griseofulvin - pharmacological characterization and therapeutic application, factors determining its biosynthesis, as well as new therapeutic opportunities. Clinical data for the latter are not provided, and the conclusions drawn are based on docking and experimental data. I recommend revising the title.

The title is changed to “Griseofulvin: An Updated Overview of Old and Current Knowledge”.

Round 2

Reviewer 3 Report

Dear Authors,

In the new version of your article, you answered all my recommendations and remarks. I recommend that it be published in its current form.

Author Response

Thank you for your time and comments to improve the content of the manuscript!